# Overview of Neutralizing Antibodies and Their Potential in COVID-19

**DOI:** 10.3390/vaccines9121376

**Published:** 2021-11-23

**Authors:** José Javier Morales-Núñez, José Francisco Muñoz-Valle, Paola Carolina Torres-Hernández, Jorge Hernández-Bello

**Affiliations:** 1Institute of Research in Biomedical Sciences, University Center of Health Sciences (CUCS), University of Guadalajara, Guadalajara 44340, Mexico; jose.morales2599@alumnos.udg.mx (J.J.M.-N.); drjosefranciscomv@cucs.udg.mx (J.F.M.-V.); 2Immunology Laboratory, University Center of Health Sciences (CUCS), University of Guadalajara, Guadalajara 44340, Mexico; paola.torres3435@alumnos.udg.mx

**Keywords:** neutralizing antibodies, SARS-CoV-2, neutralization assays, COVID-19, monoclonal antibodies

## Abstract

The antibody response to respiratory syndrome coronavirus 2 (SARS-CoV-2) has been a major focus of COVID-19 research due to its clinical relevance and importance in vaccine and therapeutic development. Neutralizing antibody (NAb) evaluations are useful for the determination of individual or herd immunity against SARS-CoV-2, vaccine efficacy, and humoral protective response longevity, as well as supporting donor selection criteria for convalescent plasma therapy. In the current manuscript, we review the essential concepts of NAbs, examining their concept, mechanisms of action, production, and the techniques used for their detection; as well as presenting an overview of the clinical use of antibodies in COVID-19.

## 1. Introduction

SARS-CoV-2 is a newly identified coronavirus causing pneumonia-associated respiratory syndrome across the world. This disease was named coronavirus disease 2019 (COVID-19) by the World Health Organization (WHO) and, in March 2020, was declared a pandemic [1].

A total of 235,175,106 cases and 4,806,841 deaths have been confirmed globally as of 5 October 2021 [2]. According to the WHO’s weekly report, there has been an increase in cases, with under 3.1 million new cases reported last week (27 September to 3 October). On the other hand, the number of deaths has been decreasing globally [3]. It may be possible to attribute the increasing cases to the virus variants emerging in various parts of the world, while the decrease in the number of deaths is one of the more evident positive results of global vaccination campaigns.

Several approaches are currently under clinical trials to control SARS-CoV-2, including vaccines, biologic therapy with monoclonal antibodies, and convalescent plasma [4,5,6]. All of these are based on the use or induction of antibodies capable of preventing infection by blocking a step in the viral replicative cycle before the first virus-directed synthetic event; these antibodies are defined as NAbs. Many NAbs can block the binding of the S protein with ACE2, but there is the possibility that other NAbs could bind to another protein of the virus [6].

Antibody responses to T-dependent antigens are generated in germinal centers (GCs) within lymphoid tissue after antigen-primed B and T cell interactions to promote B cell differentiation, somatic hypermutation, and class switch recombination for transformation into memory B cells and plasma cells [7].

The role of antibodies in host protection against viral infections has been amply demonstrated. Antibodies neutralize viral infection or replication by targeting viral glycoproteins of enveloped viruses (such as the SARS-CoV-2 Spike (S) protein) or the protein shell of nonenveloped viruses. These proteins bind to cellular receptors and cellular membranes and mediate the viral fusion and penetration into the cytosol, respectively [8,9,10]. Nevertheless, there is the possibility of an unfavorable effect of these antibodies, where the binding of specific antibodies to the virus may promote viral invasion in specific cell types and enhance viral infection; this is known as antibody-dependent enhancement (ADE) [11].

The Coronavirus Antibody Database (CoV-AbDab) [12] has documented 1235 published/patented binding antibodies and nanobodies of SARS-CoV2 as of 1 September 2021.

The following sections review fundamental concepts regarding NAbs and neutralization tests. We also discuss the importance of antibodies as therapeutic molecules, as well as the potential of NAbs in evaluating immunity against COVID-19 after natural infection or vaccination.

## 2. Definition of Neutralizing Antibody

An optimal immune response against viruses depends on different functions of antibodies: (1) effector functions aimed at the elimination of infected cells, (2) improvement in the response of the host’s endogenous antiviral immunity, and (3) virus neutralization, preventing initial infection and viral spread [13,14,15].

There are numerous effector mechanisms of antibodies, such as antibody-dependent cell cytotoxicity (ADCC), antibody-dependent cellular phagocytosis (ADCP), and antibody-mediated complement-dependent cytotoxicity (CDC) each directed at the removal of infected cells [16,17]. Regarding point 2, antibody-bound infected cells may interact with dendritic cells to release type I interferons in order to stimulate NK cell activation [14]. In this review, we focus on defining the neutralizing role of antibodies.

NAbs could be defined as antibodies that bind to the free virus and prevent it from infecting cells [18]. Some authors (Neurath and Klame) specify that if an antibody binds to the host receptor, it cannot be deemed to be neutralization [19,20]. More detailed criteria define neutralization as the reduction in viral infectivity by binding the antibody to the surface of viral particles (virion), blocking the viral replication cycle [21,22]. 

NAbs generally block the binding of the virus to cellular receptors; however, in some cases, they may prevent conformational changes necessary for fusion of the virus with the cell membrane or proteolytic cleavage [22]. For enveloped viruses, the latest step blocked seems to be membrane fusion, i.e., entry into the cytoplasm.

Traditionally, the function of NAbs is mediated by a region called fragment antigen-binding (Fab), and non-neutralizing antibodies exert their effect near the crystallizable region (Fc). Figure 1 [14,23] illustrates this, using SARS-CoV-2 as an example.

During viral infections, individuals rarely produce broadly neutralizing antibodies (bNAbs), rare molecular entities, which must also be discussed [24]. These types of antibodies are well known in HIV-1 infection, and they are defined as an antibody capable of neutralizing most strains of an antigenic variable pathogen [25].

For the generation of bNAbs, various factors are needed. It is possible that the main one is persistent antigenic stimulation, independent of the viral load and duration of the infection; however, these two factors ensure a constant antigenic stimulation; however, other factors include viral load, features of the pathogen, and duration of infection [26]. Another critical parameter is antigenic stimulation with pathogen diversity. However, it has also been made clear that superinfection does not guarantee the development of bNAbs [27]. The highest frequency of bNAb activity is observed after approximately three years of infection in HIV patients [28]. Structurally they show a high degree of somatic mutation, suggesting that they undergo multiple maturation rounds by affinity to acquire amplitude [29].

## 3. Mechanisms of Action of Neutralizing Antibodies

Figure 2 shows a schematic representation of possible neutralization mechanisms using the interaction of SARS-CoV-2 with its receptors as an example.

SARS-CoV-2 attaches to the host cell with the aid of the spike (S) glycoprotein present on its envelope. S glycoprotein is composed of two subunits (S1 and S2) that have to be cleaved to allow for viral fusion with the cell membrane, entry into the cell, and initiation of the replication process [30]. Transmembrane serine protease 2 (TMPRSS2) or endosomal cysteine proteases cathepsins B (CTSL) and L (CTSB) perform this excision [31,32]. Protease cleavage at the S2′ site frees the fusion peptide from the new S2 N-terminal region. This fusion peptide is inserted into the host membrane and facilitates the pulling of the viral and host cell membrane into close proximity, leading to membrane fusion [33].

Angiotensin-converting enzyme 2 (ACE2), an enzyme located on the outer surface of a wide variety of cells, is the primary host cell target of the receptor-binding domain (RBD) of the S1 subunit [34]. This suggests that disruption of the RBD–ACE2 interaction would block SARS-CoV-2 cell entry; therefore, RBD has been suggested as the main target of NAbs against SARS-CoV-2 [35].

Neutralization mechanisms (Figure 2B) refer to the early step in the viral replication cycle being blocked [20,36]. The enveloped viruses enter the host cell by binding to the receptor on the cell surface. In contrast, nonenveloped viruses enter through cell membrane lysis or by creating pores in the membrane. Fusing the virus (nonenveloped/enveloped) with the host cell membrane requires particular conformational changes in the viral protein that a low pH can cause in endosomes [37,38,39].

Neutralization can be achieved through four main mechanisms: (1) NAbs binding to viral surface proteins and blocking their interaction with the host cell receptor and infection [6,40,41]; (2) NAbs binding to viral protein epitopes that interact with host cell coreceptors that are key for viral infection [20,42,43]; (3) NAbs binding to viral epitopes that are not essential for host cell receptor binding but are necessary for conformational changes needed for membrane fusion [44,45]. A variant of this mechanism would be the capability of NAbs to bind to proteins essential for host cell receptor binding, but NAbs are bound to distal epitopes of fusogenic proteins (internalized), preventing complete fusion [21].

The above mechanisms can be classified as inhibition of the entry of the virus into host cells; other mechanisms may provide post-binding inhibition. Some viruses require internalization and a pH decrease to trigger conformational changes and to perform viral membrane fusion [37,46]. The fourth mechanism of neutralization can occur once the virus is inside endosomes; the junction of the antibodies to viral surface proteins inhibits the necessary changes for the fusion of the viral membrane, causing neutralization. This last mechanism could target enveloped and naked virus particles, and it is a post-internalization neutralization [9,10].

To date, it is not clear whether all of the described mechanisms of neutralization occur in all viruses, but this is most likely not the case; indeed, the activated mechanism will depend on the viral protein target, whether it is an enveloped or nonenveloped virus.

In some cases, viruses can escape NAbs responses. However, the Fc fragments of the antibodies exert mechanisms that help the elimination of the virus, such as ADCC, ADCP, and CDC [47], which are effector mechanisms rather than neutralization mechanisms, as some authors state. These mechanisms also require Fc interaction with Fc receptors present on the surface of some immune cells: FcγR for IgG, FcαR for IgA, and FcεR for IgE [48].

## 4. Generation and Characteristics of a Neutralizing Antibody

This section focuses on two controversial questions: What type of B cells produce NAbs? Is there an isotype that predominates on NAbs?

A characteristic of humoral immunity is the production of antibodies, whose affinity for antigen develops during the immune response; this is known as affinity maturation [49]. 

Affinity maturation is based on the somatic mutation of the germline genes of immunoglobulin, a process called somatic hypermutation (SHM), which consists of point mutations that are performed by a protein called activation-induced cytosine deaminase (AID), resulting in a greater affinity of antibodies or, in the worst cases, a decrease in affinity [50,51]. This process is carried out in germinal centers (GCs), specialized microstructures formed in secondary lymphoid tissues upon infection or immunization, producing long-lived plasma cells and memory B cells, which protect against reinfection [52].

GCs are organized into two regions, the dark zone (DZ) and the light zone (LZ) [53]. In the DZ, there are B cells with a high proliferation rate (called centroblasts) and SHM. [50,54]. The centroblasts then enter the LZ (now are called centrocytes), where they capture and process antigens present on follicular dendritic cells (FDC) [55,56] and they subsequently present antigenic peptides to T follicular helper (Tfh) cells to in order receive critical survival signals and undergo selection [57,58]. Moreover, FDCs can have the ability to retain the native antigen to carry out the previous process, in addition to producing cytokines, such as BAFF, that help in the survival of the B cell [59].

In the LZ class-switch recombination (CSR) also occurs, where the constant region of the heavy chain of the antibody is changed, allowing B cells to produce IgG antibodies, IgA or IgE. This process diversifies the effector functions of antibodies, e.g., IgG can activate NK cells and phagocytes to eliminate cells infected by pathogens [60,61].

In the final stage of the germinal center process, the centrocytes exit the GC as memory B cells or high-affinity antibody-secreting plasma cells (47), but the molecular mechanisms that underlie this process are not yet entirely clear. In the case of a plasma cell, the key is the activation of the *Blimp1* master regulator (Prdm1), which, among many other functions, helps to stop the expression of transcription factors, such as *Pax5* and *Bcl6*, responsible for the maintenance of B cells and the germinal center [62,63]. For memory B cells, there is no established master regulator; however, a greater expression of *Bach2* is fundamental [64,65].

Plasma cells reside in the bone marrow and constitutively secrete antibodies; they do not possess BCR receptors, are not reactivated with antigenic re-exposition, and are responsible for producing serum antibodies that can last years after infection or vaccination [66,67,68]. On the other hand, memory B cells express BCR but do not secrete antibodies constitutively. When they meet again with the antigen, they can reactivate and form GCs to produce antibodies with a greater affinity; moreover, these cells can give rise to plasma cells and reside in circulation or peripheral lymphoid tissue [69,70,71]. The development of B cells can be seen in Figure 3.

Purtha et al. demonstrated through experiments after a viral resolution in mice that antibodies derived from long-lived plasma cells (LLPCs) were specific for a single dominant neutralizing epitope and, thus, can inhibit reinfections with the same pathogens but not for variants of the original pathogens. However, class-switched memory B cells can respond to variants that escape the neutralization of antibodies produced by LLPCs [72]. Similar findings were reported in humans after SARS-CoV-2 infection [73].

Based on the above study, NAbs can come from both populations of B cells, but it is most probably that bNAbs come from evolved memory B cells recruited into the plasma cell compartment. The increase in breadth and overall potency of memory B cell antibodies could be due to shifts in the repertoire, clonal evolution, or both [73,74].

NAbs could have increased affinity for antigen compared to the corresponding naive B-cell receptors [75], but a higher affinity does not always define a higher neutralizing capacity. The required levels of SHM and affinity maturation may vary from target to target; for example, while chronic infection may result in mutation levels from their germinal genes upwards of 30%, as seen in HIV-1 broadly neutralizing antibodies (bNAbs) [76], mutations of 5–20% may provide sufficient maturation for effective neutralization [77,78], which could be more readily achieved by vaccination [75]. Rose et al. suggest that mutation levels over 20% may be difficult to achieve by vaccination; thus, they consider goal mutation levels closer to 5–15% for those NAbs targeting specific and multiple sites of vulnerability [75]. The complementarity determining regions (CDRs) with more mutations are also variables for each type of antibody depending on the virus; as an example, for HIV-1, it has been seen to have a greater effect on CDR H3 [79]. Regarding SARS-CoV-2, Graham et al. reported a low percentage of SHM in *VH* and *VL* genes (mean of 1.9% and 1.4%, respectively) following an acute infection [80].

Regarding the isotype of a NAb, it is known that it can be IgG or IgA. The isotype may depend on the tissue involved. Astronomo et al. demonstrated in mucosal HIV-1 models that IgA antibodies were no more protective than their IgG counterparts even though IgA antibody forms are more abundant in the mucosa. Furthermore, they showed that IgG1 isotype NAbs were more protective than the IgA2 isotypes, attributable in part to the greater neutralization activity of the IgG1 variants [81].

The IgG1, IgG2, IgG3, and IgG4 subclasses differ in the size of the hinge region (the position of disulfide bonds between chains and molecular weight). IgG3 has a molecular weight of 170 kDal, while the other subtypes have a molecular weight of 146 kDal [82]. IgG1 and IgG3 are usually produced in response to proteins; IgG2 and IgG4 are produced in response to carbohydrate antigens; in addition, IgG4 undergoes a process termed Fab-arm exchange (FAE), in which bi-specific, functionally monovalent antibodies are created. This contributes to the anti-inflammatory properties of IgG4 and limits its ability to form immune complexes and activate complement [83]. Subclasses differ in their ability to activate the complement or bind to and react with Fc receptors in phagocytic cells [84]. Complement activation by IgG1 and IgG3 is 40 times higher than that by IgG2 [65]. The subclass IgG4 is not able to activate the classic complement pathway. Based on this, IgG1 and IgG3 are the IgG subclasses most linked to NAb activity against enveloped viruses since they mainly target peptides derived from viral proteins. Regarding SARS-CoV-2, Kallolimath et al. showed that IgG3 exhibited an up to 50-fold superior neutralization potency compared with that of the other IgG subclasses [85].

IgA in serum is mainly monomeric and comprises approximately 90% IgA1 and 10% IgA2. In IgA2, there are generally no light-to-heavy chain disulfide bonds; rather, there is a disulfide bridge between the light chains when forming dimers [86]. A key mediator of the effector function of IgA is FcαRI; it can trigger several elimination processes by neutrophils, monocytes, eosinophils, and some macrophages and dendritic cells [87]. IgA does not activate the classic complement path. Thus, one of the main differences between IgG and IgA in terms of functions is the ability to activate the complement.

In the case of SARS-CoV-2, Sterlin et al. showed that IgA1 dominates the response of NAbs, reaching its maximum values three weeks after infection; this subclass dominance can be explained because lungs are mucosal tissue. It has been proposed that the increased flexibility and the longer hinge in IgA1 compared to IgG would be more favorable for interactions between the IgA monomer and SARS-CoV-2 trimmer [88]. Regarding IgM levels, it is essential to take into account that it is the first immunoglobulin to appear, but its actions are more directed to effector functions by its pentameric structure and can activate complement via the classical pathway by the binding of C1q to the Fc regions of these immunoglobulins [89,90]. However, IgM, through its FcμR, has a role in B cell development, maturation, and activation; humoral immune responses; host defense; and immunological tolerance [91].

## 5. Neutralizing Antibodies and SARS-CoV-2

### 5.1. Challenges in Comparing Antibody Titers

NAbs induced by vaccines or natural infection play crucial roles in controlling viral infections [92]. In SARS-CoV-2 infection, epitopes that bind to NAbs are found predominantly in the receptor-binding domain (RBD) of the viral “S” protein [93].

The development of vaccines against SARS-CoV-2 has moved at an unprecedented speed; however, vaccine developers did not have a homogeneous system to measure immune responses after vaccination, which made immunogenicity comparisons difficult. Initiatives as the CEPI Global Centralised Laboratory Network have been launched for the harmonization of immune response assessment across COVID-19 vaccine candidates. A key tool for this harmonization is the global use of an International Standard to calibrate all assays to an arbitrary unit; therefore, it was proposed that immunogenicity results must be reported as an international standard unit (IU/mL) for neutralizing antibodies [94]. This will allow for a comparison of immune responses after natural and vaccine-induced infection. The international standard is based on pooled human plasma from convalescent patients, which is lyophilized in ampoules, with an assigned unit of 250 IU/ampoule for neutralizing activity [95].

### 5.2. NAbs Induced by Natural Infection and Their Protective Role

In natural infection, most patients infected with SARS-CoV-2 develop variable titers of NAbs between days 14 and 20 post-infection [96]. Some studies have shown that most patients had detectable SARS-CoV-2 antibody responses up to 13 months after infection, giving hope that it could last even longer than predicted [97]. However, the neutralizing antibody levels begin to decline after roughly 6–8 months, and it was estimated that ≈24% of convalescent donors lost NAbs after 6–8 months from initial symptoms [98].

Other studies have been reported that SARS-CoV-2 infection elicits robust neutralizing antibody titers that last beyond six months in most individuals [96]. However, Marot et al. showed evidence that neutralizing antibody levels begin to fall from 2 to 3 months after infection [99].

Seow et al. showed that the kinetics of the neutralizing antibody response is typical of an acute viral infection, with declining neutralizing antibody titers observed after an initial peak, and that the magnitude of this peak is dependent on disease severity [100]. Moreover, Beltran et al. showed that higher neutralization titers are good predictors of survival in patients with severe COVID-19 [101].

Patients who have recovered from severe disease have higher NAbs than patients with mild or asymptomatic infections [101,102]. This may be as a result of prolonged stimulation of the B-cell receptor [103] or due to the high production of interferon type I (IFN-I) in the course of severe disease. IFN-I plays a vital role in the early stages of the viral immune response, and it is part of the innate response. Moreover, IFN-I induces the activation of dendritic cells and, therefore, allows these cells to present antigens to virgin CD4+ and CD8+ T cells. Activated CD4+ T cells stimulate the production of specific antibodies by B cells, while CD8+ T cells are cytolytic [104].

Regarding outpatient and asymptomatic individuals, Röltgen et al. observed that SARS-CoV-2 antibodies progressively decreased after five months post-infection [105]. Dugan et al. showed that some months after infection, there was a change in the persistence of NAbs against the S protein of SARS-CoV-2, and the rate of these NAbs began to decline. In contrast, there was a transition in the production of antibodies toward non-neutralizing viral targets, such as NP and ORF8 [106]. However, other authors have shown IgG memory B cells against S glycoprotein and RBD in the blood of COVID-19 patients; therefore, there are memory responses after a natural infection, which have the potential to be activated to rapidly-produce neutralizing antibodies on re-exposure to SARS-CoV-2 [107].

### 5.3. NAbs Induced by Vaccination

Although NAbs have been determined for all of the approved vaccines, the specific assays have varied and, thus, are not directly comparable. Most of the studies have reported a good humoral response after a few days post-vaccination, but NAbs tend to decrease over time. However, memory B cells can rapidly deploy more antibodies in a re-exposure to the virus, and this is also true for T cells, which can attack already infected cells [108].

Levin et al. studied the response and kinetics of antibodies against SARS-CoV-2 six months after receiving the second dose of mRNA BNT162b2 vaccine in 3808 patients. They observed that IgG antibodies decreased at a consistent rate, whereas the neutralizing antibody level decreased rapidly for the first 3 months with a relatively slow decrease thereafter. Although IgG antibody levels were highly correlated with neutralizing antibody titers, the regression relationship between the IgG and neutralizing antibody levels depended on the time since receipt of the second vaccine dose [109].

Gobbi et al. followed up six patients with the same vaccine at 7 months, and, overall, an-ti-SARS-CoV-2 spike RBD IgG titers and neutralizing antibody titers progressively declined during follow-up [110]. However, despite this decrease in post-vaccination neutralizing antibodies, as in the natural SARS-CoV-2 infection, vaccines induce a cellular memory. Ciabattini et al. demonstrated that the mRNA BNT162b2 vaccine elicits strong B cell immunity with spike-specific memory B cells that still persist 6 months after vaccination, playing a crucial role in the rapid response to SARS-CoV-2 virus encounter [111].

Longitudinal analysis of long-term immune protection is urgently needed; modeling of the decay of the neutralization titer after immunization predicts that a significant loss in protection from SARS-CoV-2 infection will occur, as neutralization levels decline, and that booster immunization may be required within a year [112].

One concern that threatens the NAbs induced by vaccination is the emergence of SARS-CoV-2 variants with antibody escape mutations. A correlation between neutralization titers and efficacy against some viral variants has been observed [113]. These variants have been classified by the World Health Organization (WHO) based on increased transmissibility and/or pathogenicity as variants of concern (VOC), namely; alpha (B.1.1.7), beta (B.1.351), gamma (B.1.1.248), and delta (B.1.617.2), and variants of interest (VOI), namely; epsilon (B.1.427/B.1.429), iota (B.1.526), delta plus (AY.1), and lambda (C.37) [114].

Takuya et al. compared the neutralization titers of serum antibodies from individuals immunized with three vaccines: BNT162b2, mRNA-1273, and Ad26.COV2.S. The study groups were controlled for age, comorbidities, and history of pre-vaccination. The results demonstrate a high level of cross-neutralization by antibodies elicited by BNT162b2 and mRNA-1273 on the variants but significantly decreased neutralization by those elicited by the single-dose Ad26.COV2.S [115].

On the other hand, Schmitz et al. demonstrated that there are antibodies against SARS-CoV-2 with broad neutralizing capacity, capable of facing different variants of interest, such as B.1.351, B.1.1.248, and B.1.617.2, thus checking the ability of natural infection and vaccination to cope with the variants [116].

Based on the above considerations, as well as other studies, it can be concluded that the current vaccines still provide clinical benefit against most variants of concern by reducing COVID-19 disease severity; even so, the decrease in neutralization potency remains a subject that requires further study. Therefore, it is necessary to develop interventions capable of preventing the transmission of diverse SARS-CoV-2 variants, including vaccine boosters that target these variants or technologies capable of eliciting or delivering bNAbs antibodies [117].

An ideal vaccine for COVID-19 could produce bNAbs not only against SARS-CoV-2 variants but also for all human epidemic coronaviruses (HCoV).

Therefore, the question of whether a previous HCoV infection induces cross-protection against SARS-CoV-2 arises. Song et al. measured the ability of pre-existing antibodies to neutralize SARS-CoV-2 due to previous HCoV infection (SARS-CoV-1, MERS-CoV, HCoV-HKU1, HCoV-OC43, HCoV-NL63, and HCoV-229E). That study showed that pre-existing antibodies do not cross-react with SARS-CoV-2 to neutralize them, but pre-existing memory B cells can cross-react and generate antibodies against SARS-CoV-2 more quickly [118].

### 5.4. Factors Affecting NAb Production

In a Japanese cohort, immunosuppressive medication, age, glucocorticoids, and drinking alcohol have been identified as factors predicting lower antibody titers after vaccination, whereas previous SARS-CoV-2 infection, female gender, the time between two vaccine doses, and medication for allergy were identified as factors predicting higher serum antibody titers [119]. These same associations for age and previous infection were reported in another study [120].

Levin et al. reported that six months after receipt of the second dose, neutralizing antibody titers were lower among persons 65 years of age or older than among those 18 to less than 45 years of age (ratio of means, 0.58; 95% confidence interval (CI), 0.48 to 0.70), substantially lower among men than among women (ratio of means, 0.64; 95% (CI), 0.55 to 0.75), and lower among participants with immunosuppression than among those without immunosuppression (ratio of means, 0.30; 95% CI, 0.20 to 0.46) [109].

Comorbidities, such as diabetes, obesity, hypertension, dermatitis, and being overweight, have not been associated with seronegativity or low production of neutralizing antibodies [121], but kidney and liver diseases have been associated with a lower humoral response. It is possible that this poor response is linked to alterations in the immune system in renal disease, as uremia is associated with a state of immune dysfunction characterized by immunodepression [122].

Regarding natural infection, Gozalbo-Rovira et al. reported weak correlations between antibody levels and inflammatory biomarkers (ferritin, D-dimer, CRP, lactate dehydrogenase (LDH), and interleukin-6) [123,124]. These correlations could be explained by the relationship between the magnitude of antibody response with the hyperactivation of the immune system in patients with severe COVID-19. It is believed that the cytokine storm plays a key role in disease progression and thus in COVID-19 prognosis [125]. Therefore, NAb levels in recovered COVID-19 patients are positively linked with the severity of the disease [126].

## 6. Neutralization Assays

Multiple serological tests are used to evaluate virus-antibody interactions (hemagglutination inhibition test, complement fixation test, fluorescent antibody test, etc.). Still, only a few assays, such as the plaque reduction neutralization test (PRNT), measure the virus neutralization during the process of viral attachment and the entry to host cells [127,128].

The PRNT allows one to measure the effects of antibodies on viral infectivity by plating the virus with susceptible cells, as shown in Figure 4A. Thus, it is considered the gold standard to evaluate the neutralization capacity of antibodies against SARS-CoV-2. The cells are cultured in semi-solid media that restrict the spread of the virus. Each virus that initiates infection produces a localized area of infection, known as plaque, which can be visualized and counted. After the count of plaques, it is possible to determine the percent reduction in total virus infectivity [127,128,129]. This assay has a considerable disadvantage, which is usually performed with live viruses, and it is necessary to work in a biosafety level 3 (BSL-3) laboratory in the hands of very skilled and experienced people [130,131].

Virus neutralization tests (VNT) can also be performed with viral vectors pseudotyped with the S protein of SARS-CoV-2. This technique does not require a BSL-3 laboratory; still, it requires a specialized laboratory setup, and it is a very complicated and time-consuming procedure [132,133], as shown in Figure 4B.

Since the SARS-CoV-2 S protein is necessary for the virus to enter a cell, it is possible to transfect specific cells that can produce and express spike-pseudotyped lentiviral particles that can be used to infect susceptible cells that express the ACE2 receptor. Such pseudotyping can be achieved using human immunodeficiency virus (HIV)-based lentiviral particles [134,135] and vesicular stomatitis virus (VSV) [136,137,138]. These pseudotyped particles can be used to measure spike-mediated cell entry with fluorescent or luciferase reporters to evaluate the neutralizing capacity of human antibodies. Compared to live virus assays, pseudovirus-based neutralization assays (PBNA) are less laborious, as the data are obtained through luminescent reading. In contrast, live virus assays require reading the results manually in the microscope [137]. The neutralization is measured by reducing luminescence in relative light units (RLUs) [137,139].

Another approach is to evaluate neutralization without using a live virus, which is the recombinant replication VSV (rVSVs). In comparison with pseudotypes, the recombinant VSVs encoding the S protein are easier to produce. rVSVs have previously been used with other lethal viruses as well as with SARS-CoV and MERS-CoV. In this case, the native glycoprotein gene of the VSV genome is replaced with the gene encoding the S protein. The VSV genome is also modified to express a green fluorescent protein (GFP) that functions as a reporter, allowing the evaluation of infection [140,141]. Another assay for neutralization is the Immuno-Cov^TM^ developed by Vandergaast et al. [142], in which a VSV is modified to express the S protein, but a dual split protein (DSP) luciferase system is used to quantify the virus neutralization. Their DSP system uses a chimeric split green fluorescent protein (GFP) and split Renilla luciferase (RL) [143]. The fusion between two cell lines expressing complementary pieces of the reporter system allows for the virus-induced cell fusion to be measured in a 96-well plate format.

Finally, an innovative surrogate virus neutralization test (sVNT), which is based on the principle of blocking enzyme-linked immunosorbent assay (ELISA) mimicking virus-cell interaction to detect the presence of NAbs in a sample, has been developed, as shown in Figure 4C. For this, human ACE2 protein (hACE2) is immobilized in the plate, and for detection, a horseradish peroxidase (HRP)-conjugated RBD is used. Assuming there is a high presence of NAbs, there will be lower signal intensity in that case. On the contrary, if there are no NAbs, the HRP-conjugated RBD binds to the hACE2, and the signal is higher [144]. This assay is the first of its kind to be approved by the U.S. Food and Drug Administration (FDA) for diagnostic use and could be a strategy within reach of most clinical laboratories.

## 7. Therapeutic Applications of Neutralizing Monoclonal Antibodies: Can We Take Advantage of This for COVID-19?

Neutralizing monoclonal antibodies had been used against the respiratory syncytial virus (RSV) [145] and Ebola virus disease (EVD) [146,147]. To date, there are some promising results of neutralizing monoclonal antibodies in neutralization assays for in vitro models and diminutions in viral loads in the respiratory tract in animal models and some patients with COVID-19 [148,149]. We summarize the most promising approaches below. The general mechanism is illustrated in Figure 5.

### 7.1. LY-CoV555

This neutralizing monoclonal antibody (also known as LY3819253) binds with high affinity to the RBD of SARS-CoV-2. It was developed from the plasma of a convalescent patient of COVID-19 (antibody developed by Eli Lilly), and it showed passive protection in non-human primates [150]. Other authors analyzed the benefits of this monoclonal antibody in outpatients with COVID-19, with one or more mild or moderate symptoms [151]. The antibody was administrated intravenously three days after positive results for SARS-CoV-2. The results between four groups LY-CoV555, with doses of 700, 2800, 7000 mg, and a placebo group, were evaluated. The percentage of patients hospitalized with COVID-19 was 1.6% in the LY-CoV555 group and 6.3% in the placebo group. Moreover, the LY-CoV555 group had a lower symptom burden [151].

### 7.2. VIR-7831 and VIR-7832

Human monoclonal antibodies have been developed by Vir Biotechnology and GSK, and they are derived from an antibody (S309) isolated from a patient who recovered from SARS in 2003, but S309 also neutralizes SARS-CoV-2. This antibody binds to a conserved epitope shared by the two coronaviruses, diminishing the probability of mutational escape. Both mAbs contain a mutation that prolongs serum half-life and enhances distribution in the respiratory mucosa. In the case of VIR-7832, this antibody has another mutation that evokes CD8^+^ T-cell responses. These antibodies have shown neutralization of wild-type SARS-CoV-2 and variants with mutations in the S protein, such as B.1.1.7, B.1.351, and P.1. Moreover, in a Syrian Golden hamster proof-of-concept wild-type SARS-CoV-2 infection model, animals treated with VIR-7831 had less weight loss and significantly decreased total viral load and infectious virus levels in the lung compared to a control mAb [152]. These data suggest that VIR-7831 and VIR-7832 are promising new agents in the fight against COVID-19.

### 7.3. BGB-DXP593

This is a monoclonal antibody developed by BeiGene, and it was identified by the high-throughput single-cell sequencing of convalescent samples from patients who had recovered from COVID-19. This antibody probably inhibits the entrance of the SARS-CoV-2 because the antibody’s epitope overlaps with the ACE2 binding site of the S protein [153], but the exact mechanism has not been fully elucidated. A phase 2 trial of BGB-DXP593, which included 181 participants and involved evaluating its efficacy and safety in patients with mild to moderate COVID-19, was carried out, but there are still no publications of the results (ClinicalTrials.gov identifier: NCT04551898).

### 7.4. REGN-COV2

This is a cocktail of two SARS-CoV-2 NAbs (REGN10987+REGN10933). These antibodies are directed to the S protein to prevent viral entry into human cells through the ACE2 receptor [154,155]. The interim analysis of the ongoing trial for REGN-COV2 (ClinicalTrials.gov, NCT04425629) showed a diminished viral load in nonhospitalized patients, with a more significant effect in individuals whose immune response had not been initiated (serum antibody negative) and patients with higher viral loads at baseline. In terms of safety, the adverse effects were similar in the REGN-COV2 and the placebo groups [156].

Currently, another phase 3 study is evaluating the efficacy of REGN-COV2 in preventing asymptomatic or symptomatic SARS-CoV-2 infection (Clinical-Trials.gov, NCT04452318), in which 3,750 participants are enrolled, including pediatric subjects (<12 years) [156].

### 7.5. CT-P59

This is another monoclonal antibody, which blocks the interaction between SARS-CoV-2 and ACE2 receptors via steric hindrance. This antibody has been evaluated in animal models with ferrets, golden Syrian hamsters, and rhesus monkeys. These models have demonstrated a reduction in viral loads and improved clinical symptoms and lung pathology [157]. CT-P59 is currently under a phase 1 trial (Clinical-trials.gov, NCT04593641) with 18 enrolled individuals, and the trial´s goal is to evaluate the safety, tolerability, and virology of CT-P59 in patients with mild symptoms of SARS-CoV-2 infection. Furthermore, other trials are being conducted, with one evaluating the effects of CT-P59 on mild and moderate symptoms, with 1,020 subjects enrolled (Clinical-trials.gov, NCT04602000), and another determining the safety, tolerability, and pharmacokinetics in healthy subjects (Clinical-trials.gov, NCT04525079).

### 7.6. Nanobodies

Another innovative and promising approach with variable domains of heavy-chain-only (VHH) antibodies is nanobodies (Nbs). These are small (~15 kD) monomeric antigen-binding domains derived from single-chain antibodies [158]. Nbs are very appealing therapeutic agents because of their physicochemical properties, which permit them to be administered by inhalation, minimizing the doses and probably the adverse effects [159,160].

A multivalent Nb directed to the RBD of the SARS-CoV-2 S protein has demonstrated very potent neutralization in PRNT assays [161]. Other Nbs were identified by phage display using nanobody libraries from an alpaca and a llama immunized with the RBD and inactivated virus. Four nanobodies, named VHHs E, U, V, and W, potently neutralized SARS-CoV-2 and SARS-CoV-2 pseudotyped vesicular stomatitis virus. These nanobodies were also used in combinations; the VE combination triggered the premature induction of postfusion conformation, which irreversibly inactivates the S protein [162].

## 8. Disadvantages of Neutralizing Antibodies: Antibody-Dependent Enhancement (ADE)

Care should be taken in the use of NAbs as therapeutics or their induction via vaccination so as not to cause adverse side effects, such as antibody-dependent enhancement (ADE). The antiviral activity exerted by the antibodies is mediated by the direct inhibition of the entrance of the virus to host cells (neutralization) and by effector functions [163]. However, in some viruses, the binding of specific antibodies to viral surface proteins may promote viral invasion in specific cell types and enhance viral infection [11]; this effect is called ADE.

The phenomenon of ADE is mainly related to the effector functions of antibodies, called non-neutralizing antibodies (nnAbs) [164]. In the case of HIV-1, it is clear that its role in the development of the disease occurs at the beginning of the course of infection, typically directed against gp41 [165]. Willey et al. demonstrated the involvement of the complement in HIV in the development of ADE. The high levels of enhancement seen through CR2 (Complement receptor type 2) may reflect an important role for nnAbs [165]. 

ADE can occur via two main mechanisms: dependent of Fc receptor (FcR) or complement [166]. The first mechanism of ADE has been observed to be involved in the immunopathogenesis of severe dengue forms, and it has also been found in West Nile virus and HIV [166,167]. This mechanism depends on FcR, an antibody receptor targeting the Fc portions on antibodies located at the membrane of certain immune cells, including B lymphocytes, NK cells, macrophages, neutrophils, and mast cells [168]. In this mechanism, after the antibodies have attached to a viral protein, this virus-antibody immunocomplex strengthens viral adhesion through the interaction of the Fc portion on the antibody and its receptor on the surface of particular cells [166,169].

The second mechanism could be C3 or C1q dependent. Regarding C3, it is activated in the classical pathway through the virus-antibody immunocomplex. Then, the C3-corresponding receptor interaction enhances viral adhesion in the form of the virus–antibody–complement complex [166]. For the C1q-dependent mechanism, virus–antibody C1q complexes promote fusion between the viral capsule and the cell membrane by the deposition of C1q and its receptor. This complex binds to the C1q receptor in cells, initiates the intracellular signaling pathway, and then promotes the virus-specific receptor attached, as well as endocytosis, of the target cells [170].

NK cells identify IgG-viral protein complexes in infected cells via FcγR to mediate antibody-dependent cytotoxicity. Myeloid cells use these interactions to eliminate opsonized virions and virus-infected cells by antibody-dependent cellular phagocytosis [171]. The pathway of the complement is also activated by binding Fc to the C1q component of the complement, resulting in the opsonization of viruses or infected cells and the recruitment of myeloid cells [11]. Both cases of ADE can occur when non-neutralizing antibodies or antibodies with sub-neutralizing capacity bind to viral antigens without blocking [166].

Antibodies mediating FcR and complement-dependent effector functions may or may not have neutralizing activity [172,173].

Dengue is an example of the low neutralization potential of the antibodies by which the ADE phenomenon develops. The decrease in antibody titers as time passes between primary infection and when a heterologous infection is contracted secondary can explain this phenomenon in dengue. For this reason, it has been established that after a period of two or more years between these two events is when the phenomenon of ADE can be presented with greater probability [174].

Based on the above, the following question arises: can vaccines against COVID-19 lead to intensified ADE infections in humans? This is unlikely, because human coronavirus diseases lack the biological or pathological attributes of dengue virus (DENV) ADE disease. Unlike SARS-CoV and MERS-CoV, DENV predominantly infects monocytes, macrophages, and dendritic cells and not the respiratory epithelium. Those phagocytic cells abundantly express both viral entry receptors and FcγRs [172]. The mechanism of ADE of disease is associated with dengue; therefore, it depends mainly on the capacity of DENV to infect FcγR-expressing myeloid cells and sequential infection of the same person with different viral serotypes (11).

On the other hand, Yunjiao Zhou et al. measured the antibody-dependent entry of SARS-CoV-2 pseudoviruses in the presence of monoclonal antibodies. Eleven of forty-eight antibodies (23%) significantly enhanced viral infection of Raji cells, while no viral infection was induced by the presence of different concentrations of human serum samples collected from healthy donors or convalescent individuals antibodies of convalescences patients [175]. Thus, the determination of the ADE effect would be a crucial step for the clinical use of potent monoclonal NAbs.

## 9. Conclusions

Antibody production is part of an efficient immune response against SARS-CoV-2, and some antibodies contribute to effector functions to eliminate the infectious agent, while others, named NAbs, can neutralize the virus. Most of these antibodies can bind to the S protein of the virus, preventing it from binding to the host’s ACE2, thereby preventing infection.

NAbs have become a subject of relevance during the COVID-19 pandemic since the quantification of these antibodies has made it possible to evaluate, to some extent, the immunity generated against SARS-CoV-2, either by natural infection or by the different vaccines developed for this disease. In addition, their ability to prevent or reduce the infectivity of the virus makes them promising therapeutic tools for COVID-19. There have been encouraging results in animal and experimental models with monoclonal NAbs, but it is necessary to wait for the results of clinical trials to confirm their therapeutic utility. Multiple clinical trials are currently underway to test different monoclonal NAbs for prophylactic and therapeutic purposes.

We must not lose sight of SARS-CoV-2 mutations, especially those associated with a viral escape to NAbs. Therefore, it is essential to evaluate the capacity of these monoclonal antibodies against the variants that are currently being observed globally to determine the need to develop new antibodies that are effective against them. As is well known, the virus will continue to mutate, so it is necessary to focus efforts on the development of monoclonal antibodies directed against highly conserved epitopes so that they are not prone to mutations and to ensure that antibodies do not lose effectiveness against the majority of the emerging variants.

## Figures and Tables

**Figure 1 vaccines-09-01376-f001:**
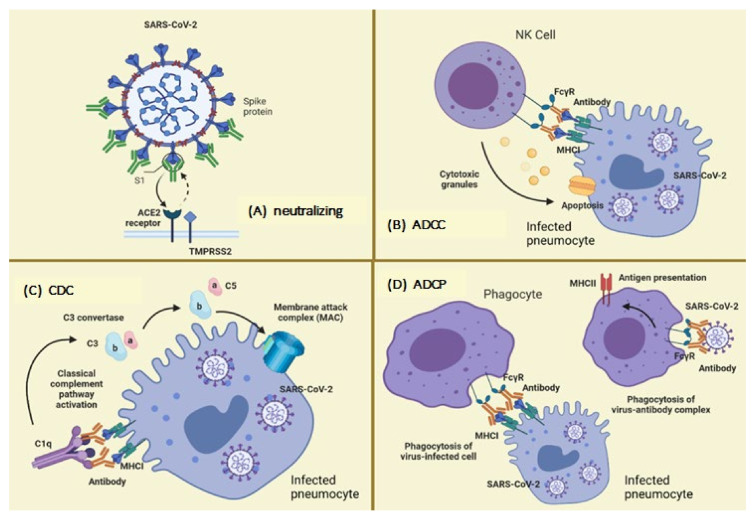
Neutralizing and non-neutralizing actions of antibodies. (**A**) The neutralizing action is carried out through the variable fraction (Fab) of the antibody, whose primary limitation is viral resistance. Here, effector mechanisms enter to avoid viral replication. For example, in the ADCC mechanism (**B**), Fc gamma receptors present in some cells (e.g., natural killers cells, NK) engage with antibody-bound infected cells and induce target cell death through the release of cytotoxic granules; in the CDC mechanism (**C**), binding of C1q to antibody-bound virus-infected cells leads to the activation of the classical complement pathway, or the ADCP mechanism (**D**), phagocytes can clear virus-infected cells and immune complexes that are engaged by Fc gamma receptors through phagocytosis. Abbreviations: ADCC, antibody-dependent cell cytotoxicity; CDC antibody-mediated complement-dependent cytotoxicity; ADCP, antibody-dependent cellular phagocytosis; ACE2, angiotensin-converting enzyme 2; TMPRSS2, transmembrane serine protease 2.

**Figure 2 vaccines-09-01376-f002:**
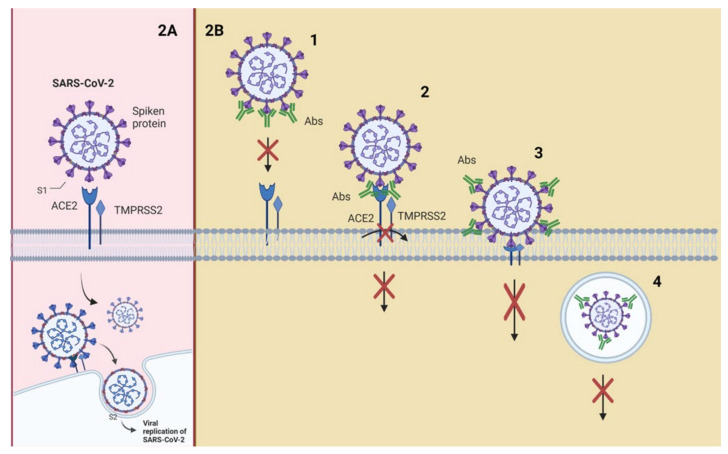
Interaction of SARS-CoV-2 with its receptors and neutralization mechanisms. (**A**) S1 contains the receptor-binding domain (RBD) and directly binds to ACE2 to gain entry into host cells. (**B**) Neutralizing mechanisms: (1) NAbs bound to the receptor-binding protein (S) and block its interaction with ACE2; (2) the virion establishes contact between its binding protein and the receptor on the cell surface, and NAbs block subsequent steps, such as binding to a coreceptor; (3) the virion is about to fuse with the cell membrane, but NAbs are bound to proteins that are not essential for cell receptor binding but exert conformational changes that do not allow virus internalization with the cell membrane; (4) NAbs prevent the virion from merging its envelope with the vesicular membrane (endosome) and begin viral replication, and the binding of the antibody to the virus inhibits the conformational changes necessary for membrane fusion. Abbreviations: ACE2, angiotensin-converting enzyme 2; TMPRSS2, transmembrane serine protease 2.

**Figure 3 vaccines-09-01376-f003:**
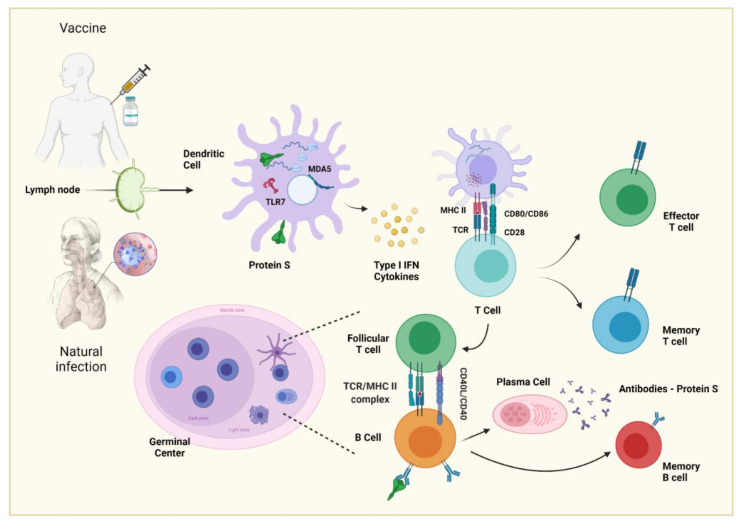
Production of neutralizing antibodies by natural infection/vaccination. The antigen is processed in a dendritic cell to ensure its presentation and to produce cytokines. The dendritic cell within the lymph node presents antigen by class II MHC to naive T-lymphocytes, which differentiate in an effector or memory T-cell. The B cell can recognize native, unprocessed antigens with its BCR, but to perform a more specific response, the antigen needs to enter a germ center to be in contact with a follicular T cell that has already been presented with it, as this allows for it to be presented to the B cell and for the germ center reaction to begin, thereby producing specific antibodies, in this case, those against SARS-CoV-2.

**Figure 4 vaccines-09-01376-f004:**
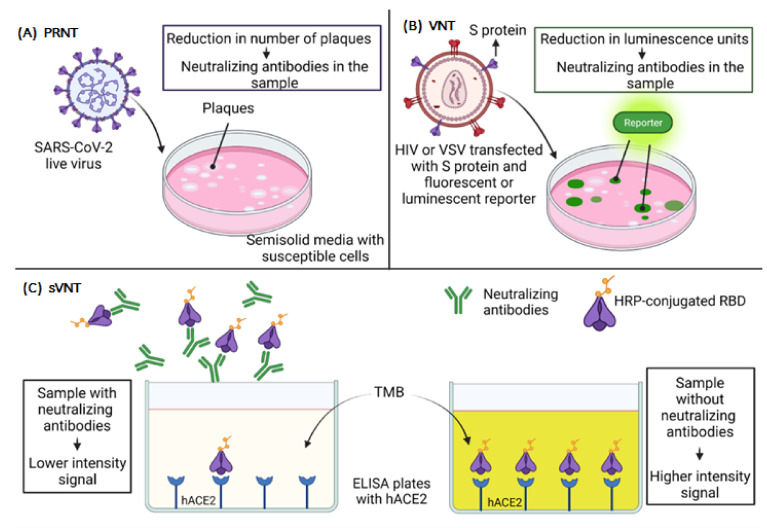
Neutralization assays. (**A**) In a PRNT assay, cells susceptible to infection are cultured in semi-solid media and are infected with the SARS-CoV-2 live virus. The semi-solid media allow that the infection to be localized and can be visualized as plaques. The plaques are counted, and a reduction in the number of plaques means that there are neutralizing antibodies in the sample. (**B**) VNT is similar to a PRNT, but instead of a live virus, a virus-like HIV or VSV is transfected with the S protein of SARS-CoV-2 and a luminescent reporter. The luminescence units are sites of infection, and a reduction in luminescence units means that neutralizing antibodies are present. (**C**) The sVNT uses an immobilized hACE2 in ELISA plates and an HRP-conjugated RBD; if neutralizing antibodies are present in the sample, the HRP-conjugated RBD will not bind to hACE2, and there will be no or lower intensity signal. PRNT, plaque reduction neutralization test; VNT, virus neutralization test; sVNT, surrogate virus neutralization test; HIV, human immunodeficiency virus; VSV, vesicular stomatitis virus; hACE2, human angiotensin-converting enzyme 2; HRP, horseradish peroxidase; RBD, receptor-binding domain; TMB, tetramethylbenzidine.

**Figure 5 vaccines-09-01376-f005:**
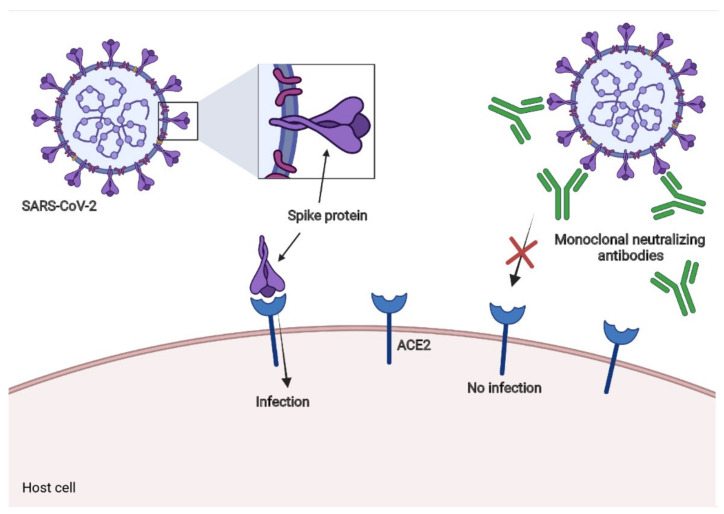
Monoclonal neutralizing antibodies. The general mechanism of most of the monoclonal antibodies mentioned here is that of binding to the S protein; in this way, these antibodies could prevent SARS-CoV-2 from attaching to ACE2 and, therefore, decrease the infectivity.

## Data Availability

The data that support the findings of this study are available on request from the corresponding author.

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
