# Peer review of "Overview of Neutralizing Antibodies and Their Potential in COVID-19"

_vaccines, 2021, doi:10.3390/vaccines9121376_

Round 1
Reviewer 1 Report
Review
Overall, this review is comprehensive, clear, well laid out and a very useful review of a topic of intense interest. The manuscript would benefit from review from a copy editor to address minor errors of idiom and syntax.
I would suggest that the authors rewrite section 5 to cover clinical aspects of neutralising antibodies and SARS-CoV-2 as follows. I’ve suggested some additional references which the authors could use in addition to those already in the review.
- Challenges in comparing antibody titres, and interpreting correlation between titres and protection
- WHO International Standard for anti-SARS-CoV-2 immunoglobulin. PA Kristiansen, M Page, V Bernasconi, G Mattiuzzo… - The Lancet, 2021
- Krammer, F. A correlate of protection for SARS-CoV-2 vaccines is urgently needed. Nat Med 27, 1147–1148 (2021). https://doi.org/10.1038/s41591-021-01432-4
- Nabs induced by natural infection
- Nature
- Duration/longitudinal evolution
- Host factors that affect Nab titres
- Solodky ML, Galvez C, Russias B, et al. Lower detection rates of SARS-COV2 antibodies in cancer patients versus health care workers after symptomatic COVID-19. Ann Oncol. 2020;31(8):1087-1088. doi:10.1016/j.annonc.2020.04.47
- Disease factors that affect NAbs
- Timing/severity
- Seow, J., Graham, C., Merrick, B. et al. Longitudinal observation and decline of neutralizing antibody responses in the three months following SARS-CoV-2 infection in humans. Nat Microbiol 5, 1598–1607 (2020). https://doi.org/10.1038/s41564-020-00813-8
- Defining the features and duration of antibody responses to SARS-CoV-2 infection associated with disease severity and outcome, Roltgen et al. Science Immunology • 18 Dec 2020 • Vol 5, Issue 54DOI: 10.1126/sciimmunol.abe0240
- COVID-19-neutralizing antibodies predict disease severity and survival. Garcia-Beltran et al
- Protective function of NAbs
- quantitation of neutralization potency revealed that high potency was a predictor of survival (COVID-19-neutralizing antibodies predict disease severity and survival. Garcia-Beltran et al)
- Antibodies from vaccines
- Inter-individual differences
- Takahiro Kageyama, Kei Ikeda, Shigeru Tanaka, Toshibumi Taniguchi, Hidetoshi Igari, Yoshihiro Onouchi, Atsushi Kaneda, Kazuyuki Matsushita, Hideki Hanaoka, Taka-Aki Nakada, Seiji Ohtori, Ichiro Yoshino, Hisahiro Matsubara, Toshinori Nakayama, Koutaro Yokote, Hiroshi Nakajima. Antibody responses to BNT162b2 mRNA COVID-19 vaccine and their predictors among healthcare workers in a tertiary referral hospital in Japan. Clinical Microbiology and Infection, 2021
- Protection from vaccines, duration of vaccine responses and role of boosters
- Summarised here: https://www.nature.com/articles/d41586-021-02532-4
- Effect of mutations on vaccine efficacy
- Wilfredo F. Garcia-Beltran, Evan C. Lam, Kerri St. Denis, Adam D. Nitido, Zeidy H. Garcia, Blake M. Hauser, Jared Feldman, Maia N. Pavlovic, David J. Gregory, Mark C. Poznansky, Alex Sigal, Aaron G. Schmidt, A. John Iafrate, Vivek Naranbhai, Alejandro B. Balazs. Multiple SARS-CoV-2 variants escape neutralization by vaccine-induced humoral immunity. Cell,2021. https://doi.org/10.1016/j.cell.2021.03.013.
- Inter-individual differences
- Timing/severity
I would also suggest the authors cover the role of non-neutralising antibodies in ADE of infection in section 8 – e.g. in HIV
- Willey, S., Aasa-Chapman, M.M., O'Farrell, S. et al. Extensive complement-dependent enhancement of HIV-1 by autologous non-neutralising antibodies at early stages of infection. Retrovirology 8, 16 (2011). https://doi.org/10.1186/1742-4690-8-16
- Scott B Halstead, Suresh Mahalingam, Mary A Marovich, Sukathida Ubol, David M Mosser. Intrinsic antibody-dependent enhancement of microbial infection in macrophages: disease regulation by immune complexes. The Lancet Infectious Diseases, 2010,
Author Response
Reviewer 1
|
Specific comments |
Corrections |
|
The manuscript would benefit from review from a copy editor to address minor errors of idiom and syntax. |
The manuscript was edited by the MDPI team of first-language English speakers who improved the grammar and phrasing of our paper. |
|
I would suggest that the authors rewrite section 5 to cover clinical aspects of neutralising antibodies and SARS-CoV-2 as follows. I’ve suggested some additional references which the authors could use in addition to those already in the review. |
Section 5 was restructured as follows: 5.1 Challenges in comparing antibody titers (lines 286-300) 5.2 NAbs induced by natural infection and their protective role (lines 302-334) 5.3 NAbs induced by vaccination (lines 336-393) 5.4 Factors affecting NAb production (lines 395-420)
We added all the articles that the reviewer suggested. |
|
I would also suggest the authors cover the role of non-neutralising antibodies in ADE of infection in section 8 – e.g. in HIV |
The role of non-neutralizing antibodies in the development of ADE was included and the example of HIV was explained (582- 587). Additionally, this topic is developed in Figure 1. |
Thank you for your valuable comments.
Reviewer 2 Report
General comments
Morales-Nunez et al. have written a fine, comprehensive review of SARS-CoV-2 neutralization by antibodies, against the background of virus neutralization in general, while also including some aspects of effector functions and antibody-dependent enhancement of infection. The topic is timely and important. Because of the fast growth of the COVID-19 literature, much confusion occurs. After some clarifications as listed below, this article could become a valuable resource for the field. The figures are great.
Specific comments
38: Many but not all SARS-CoV-2-specific NAbs do indeed block the binding of the S protein to the receptor ACE2. It is not certain that there are no ancillary factors that could first anchor the virus to cells. Neutralization is not defined like that, but as the inhibition of infection by binding to the virion surface and thereby blocking a step in the viral replicative cycle before the first virus-directed synthetic event. For enveloped viruses the latest step blocked seems to be the membrane fusion, i.e., entry into the cytoplasm. The statement needs of those three corrections.
59: Point 2 requires clarification.
68-69: This is better that 38 (which it contradicts). The two definitions given should not be thought to contradict each other: they are basically the same, with a little more detailed criteria in the latter.
70: Not based on the former either!
75: Fab also has constant domains; and c in Fc does not stand for constant but crystallizable.
93: Sometimes dependent on viral load and duration, no? As it says for duration on line 97. This could be clarified.
117-122: Explain the double cleavage.
148-149: No conflict: it’s the fusion that counts as entry, not the endocytosis.
154: all those listed block entry into the cytoplasm (“host” is not specific enough). Naked viruses can be neutralized after entry.
158: Or Fc alpha receptors for IgA etc.
220: Ultimately from plasma cells, though?
224-233: Interesting to compare with percentages SHM for SARS-CoV02 NAbs.
242: Over-simplistic for IgG4.
246: Explain.
260: Again, please explain.
271: “positive correlation with sex” might be misunderstood.
320: A little harsh?
326: biological here refers to replicating as opposed to pseudotyped vius?
330: Does this only refer to SARS-CoV-2? It’s not the gold standard for many viruses today.
431: The epitopes, not the NAbs, overlap.
482: Clarify mechanism 1.
495: Do the inactivated SARS-CoV-2 vaccines really do this? Are there data to support that? These vaccines are still used.
524: Most of these antibodies, not all, as noted before.
541: Conserved epitopes.
Author Response
Thank you for your valuable comments.
Reviewer 2:
|
Specific comments |
Corrections |
|
Moderate English changes required |
The manuscript was edited by the MDPI team of first-language English speakers who improved the grammar and phrasing of our paper. |
|
38: Many but not all SARS-CoV-2-specific NAbs do indeed block the binding of the S protein to the receptor ACE2. It is not certain that there are no ancillary factors that could first anchor the virus to cells. Neutralization is not defined like that, but as the inhibition of infection by binding to the virion surface and thereby blocking a step in the viral replicative cycle before the first virus-directed synthetic event. For enveloped viruses the latest step blocked seems to be the membrane fusion, i.e., entry into the cytoplasm. The statement needs those three corrections. |
We add the three suggested corrections to that statement.
The modification is observed from lines 37 to 41. |
|
59: Point 2 requires clarification. |
We clarify point 2, lines 70-71: “Regarding point 2, antibody-bound infected cells may interact with dendritic cells to re-lease type I interferons in order to stimulate NK cell activation”. |
|
68-69: This is better that 38 (which it contradicts). The two definitions given should not be thought to contradict each other: they are basically the same, with a little more detailed criteria in the latter. |
We homogenize the definitions, 73-77: “NAbs could be defined as antibodies that bind to the free virus and prevent it from infecting cells. Some authors (Neurath and Klame) specify that if an antibody binds to the host receptor, it cannot be deemed to be neutralization. More detailed criteria define neutralization as the reduction in viral infectivity by binding the antibody to the surface of viral particles (virion), blocking the viral replication cycle.” |
|
70: Not based on the former either! |
We make the correction, lines 78-81. The paragraph was changed as follows: “NAbs generally block the binding of the virus to cellular receptors; however, in some cases, they may prevent conformational changes necessary for fusion of the virus with the cell membrane or proteolytic cleavage. For enveloped viruses, the latest step blocked seems to be membrane fusion, i.e., entry into the cytoplasm.” |
|
75: Fab also has constant domains; and c in Fc does not stand for constant but crystallizable. |
We clarify this paragraph according to the suggestions, lines 82-84: “Traditionally, the function of NAbs is mediated by a region called fragment antigen-binding (Fab), and non-neutralizing antibodies exert their effect near the crystallizable region (Fc)”. |
|
93: Sometimes dependent on viral load and duration, no? As it says for duration on line 97. This could be clarified. |
We clarify this paragraph according to the suggestions, lines 99-102: “It is possible that the main one is persistent antigenic stimulation, independent of the viral load and duration of the infection; however, these two factors ensure a constant antigenic stimulation; however, other factors include viral load, features of the pathogen, and duration of infection”. |
|
117-122: Explain the double cleavage. |
We clarify this paragraph as follows (lines 130-132): “Protease cleavage at the S2′ site frees the fusion peptide from the new S2 N-terminal region. This fusion peptide is inserted into the host membrane and facilitates the pulling of the viral and host cell membrane into close proximity, leading to membrane fusion.” |
|
148-149: No conflict: it’s the fusion that counts as viral entry, not the endocytosis. |
Thank you, we clarify this paragraph as follows (lines 156-158): The fourth mechanism of neutralization can occur once the virus is inside endosomes; the junction of the antibodies to viral surface proteins inhibits the necessary changes for the fusion of the viral membrane, causing neutralization. |
|
154: all those listed block entry into the cytoplasm (“host” is not specific enough). Naked viruses can be neutralized after entry. |
Thank you, we clarify this paragraph as follows (lines 158-163): “This last mechanism could target enveloped and naked virus particles, and it is a post-internalization neutralization [9], [10]. To date, it is not clear whether all of the described mechanisms of neutralization occur in all viruses, but this is most likely not the case; indeed, the activated mechanism will depend on the viral protein target whether it is an enveloped or non-enveloped virus.” |
|
158: Or Fc alpha receptors for IgA, etc. |
We complement the paragraph as follows (lines 167-168): ”These mechanisms also require Fc interaction with Fc receptors present on the surface of some immune cells: FcγR for IgG, FcαR for IgA, and FcεR for IgE”. |
|
220: Ultimately from plasma cells, though? |
Yes, We clarify it (lines 227- 230): “Based on the above study, NAbs can come from both populations of B cells, but it is most probably that bNAbs come from evolved memory B cells recruited into the plasma cell compartment. The increase in breadth and overall potency of memory B cell antibodies could be due to shifts in the repertoire, clonal evolution, or both”. |
|
224-233: Interesting to compare with percentages SHM for SARS-CoV02 NAbs. |
We add information about it (lines 242-244): “Regarding SARS-CoV-2, Graham et al. reported a low percentage of SHM in VH and VL genes (mean of 1.9% and 1.4%, respectively) following an acute infection”. |
|
242: Over-simplistic for IgG4. |
We complemented the information (lines 255-258): “in addition, IgG4 undergoes a process termed Fab-arm exchange (FAE), in which bi-specific, functionally monovalent antibodies are created. This contributes to the an-ti-inflammatory properties of IgG4 and limits its ability to form immune complexes and activate complement.” |
|
246: Explain. |
We explain that paragraph (lines 261-265): “Based on this, IgG1 and IgG3 are the IgG subclasses most linked to NAb activity against enveloped viruses since they target mainly peptides derived from viral proteins. Regard-ing SARS-CoV-2, Kallolimath et al. showed that IgG3 exhibited an up to 50-fold superior neutralization potency compared with that of the other IgG subclasses”. |
|
260: Again, please explain. |
We explain that paragraph (lines 279-283): “actions are more directed to effector functions by its pentameric structure and can activate complement via the classical pathway by binding of C1q to the Fc regions of these immu-noglobulins. However, IgM, through its FcμR, has a role in B cell development, maturation and activation; humoral immune responses; host defense; and immunological tolerance”. |
|
271: “positive correlation with sex” might be misunderstood. |
The idea was corrected (line 404): gender is specified as a factor that affects the production of antibodies. |
|
320: A little harsh? |
This line was removed, and section 5 was restructured at the request of reviewer 1 |
|
326: biological here refers to replicating as opposed to pseudotyped vius?. |
Yes, this term of biological virus neutralization was meant to describe the neutralization of the virus replication and infectivity. Thank you for your observation. (Line 424) |
|
330: Does this only refer to SARS-CoV-2? It’s not the gold standard for many viruses today. |
Thank you, we made the corresponding modification (Line 429). |
|
431: The epitopes, not the NAbs, overlap. |
We clarify this (line 530): This antibody probably inhibits the entrance of the SARS-CoV-2 because the antibody’s epitope overlaps with the ACE2 binding site of the S protein. |
|
482: Clarify mechanism 1. |
This was clarified (lines 588-604) |
|
495: Do the inactivated SARS-CoV-2 vaccines really do this? Are there data to support that? These vaccines are still used. |
Inactivated or attenuated vaccines generate antibodies against various antigenic proteins of SARS-CoV-2, both for protein S, N, M, etc. Structural proteins other than S have not been involved in neutralization mechanisms, as they do not break the bond with the cell receptor. As their role in this function is not proven, they could develop this phenomenon at some point. However, it has not been reported to date. Therefore, this paragraph was deleted. |
|
524: Most of these antibodies, not all, as noted before. |
Thank you, we specify this (line: 639): “Most of these antibodies… |
|
541: Conserved epitopes. |
Thank you, it was corrected ( line 656) |
Reviewer 3 Report
The manuscript submitted by Morales-Núñez et al., entitled "Overview of neutralizing antibodies and their potential in COVID-19" to the SI "Virus Neutralizing and Enhancing Antibodies" of Vaccines Journal, reviews in deep the current knowledge about immune responses target during COVID infection, revisiting the ultimated antibodies against COVID.
The manuscript is very well written and in my opinion is suitable for publication at the present form.
Author Response
Thank you for your kind comment. Now, the manuscript was edited by the MDPI team of first-language English speakers who improved the grammar and phrasing of our paper.